# The Moderating Effect of Parenting on Adaptation of Children with Leukemia



**Marta Tremolada** [1,2,*], **Livia Taverna** [3], **Sabrina Bonichini** [1], **Maria Caterina Putti** [2], **Manuela Tumino** [2] and **Alessandra Biffi** [2]

1    Department of Development and Social Psychology, University of Padua, 35131 Padua, Italy; s.bonichini@unipd.it
2    Department of Child and Woman Health, University of Padua, 35127 Padua, Italy; mariacaterina.putti@unipd.it (M.C.P.); manuela.tumino@aopd.veneto.it (M.T.); alessandra.biffi@unipd.it (A.B.)
3    Faculty of Education, Free University of Bolzano-Bozen, 39042 Bolzano, Italy; livia.taverna@unibz.it
*    Correspondence: marta.tremolada@unipd.it; Tel.: +39-3474868835

**Abstract:** Parents' attitudes and practices may support the children's reactions to treatments for leukaemia and their general adjustment. This study has two aims: to explore parenting depending on the child's age and to develop and test a model on how family processes influence the psycho-social development of children with leukaemia. Patients were 118 leukemic children and their parents recruited at the Haematology–Oncologic Clinic of the Department of Paediatrics, University of Padua. All parents were Caucasian with a mean age of 37.39 years (SD = 6.03). Children's mean age was 5.89 years (SD = 4.21). After the signature of the informed consent, the parents were interviewed using the EFI-C from which we derived Parenting dimension and three parental perceptions on the child's factors. One year later, the clinical psychologist interviewed again parents using the Vineland Adaptive Behavior Scales (VABS). The analyses revealed the presence of a significant difference in parenting by the child's age: Infants required a higher and more intensive parenting. The child's coping with medical procedures at the second week after the diagnosis, controlled for parenting effect, impacted upon the child's adaptation one-year post diagnosis. Specific intervention programmes are proposed in order to help children more at risk just after the diagnosis of developmental delays.

**Keywords:** children; leukaemia; in treatment; parenting; adaptation; moderation effect



## 1. Introduction

The main task of parenting is to give care. Caring for children involves responsibility for their well-being and knowledge about their needs and how to accommodate them. Most importantly, there is an emotional bond between parents and their children that is essential to children's development and happiness [1]. The central task for parents is not simply to keep the child alive, or provide appropriate discipline, but "create the conditions in which children can develop their fullest capacity both inside and outside the family" [2]. The responsibilities and challenges involved in childcare are numerous and parents usually find ways to manage it, even in extreme conditions, such when a child has cancer.

Having a child with cancer is an overwhelming life experience for both parents. It causes psychological distress, especially at the time of diagnosis but also throughout all the therapy cycles, that can impact on the parental health locus of control [3] and can bring to post-traumatic stress symptoms [4,5].

The responsibilities and challenges involved in childcare are numerous and parents usually find ways to manage them, even in extreme conditions, such when a child has cancer. Taking care of the child with leukaemia is physically and emotionally demanding because parents must negotiate a myriad of potentially stressful situations daily: Talking with the child about the illness, learning how to correctly perform a home care regimen,

managing the parent's own emotions about the child's disease and survival, advocating on the child's behalf within a complex health care system, providing care for unaffected siblings, and fulfilling extra-familiar obligations (employment, financial) during periods of child sickness or hospitalisation [6]. 'Being there' is described as a parental response to the perceived vulnerability of the child and the parental need to give meaning to parenthood. It serves two purposes: protection and preservation. Protection means guarding the child against the negative aspects of illness and treatment. Preservation refers to the way parents influence the child's perception of his/her life, thus contributing to his/her coping and willingness to undergo treatment, to maximise the chances for survival [7]. In this sense, parenting can be a resource that help children cope with the several daily painful procedures, in order to have also a better adaptive behaviour when the therapy cycles become less intensive, and the children can return to some normal developmental activities.

Especially for mothers, caring evokes an intense emotional interdependence with their sick child, and involves a range of technical tasks and emotional work, including acting as "brokers" of information for their child and managing the cooperation with treatment [8]. When a child has cancer, parents enter a world where the terrain is unfamiliar and their basic childrearing tasks are challenged. Problems such as overprotectiveness, difficulty with consistent discipline and expression of appropriate concerns about "spoiling" the child, can occur in parenting tasks [9]. Parents of the pre-schooler children with Acute Lymphoblastic Leukemia (ALL) reported significantly more lax parenting practices and strategies associated with their child's sleep (i. co-sleeping, comforting activities, and offering more food and drink in the bedroom than the healthy group) [10] that can contribute to children's sleeping difficulties. Additionally, parents respond to child executive function difficulties with greater overprotection, which may be adaptive but not conducive to the development of independence. Although most parents report normative levels of child vulnerability and overprotection, a small subset demonstrate parenting practices that may place some survivors at risk for adverse outcomes [11].

Adherence to paediatric cancer treatment can be difficult for families, especially when the child is a young child, and the required tasks include medical procedures (mouth care and physical exams) [12]. More important, some of these difficulties are related to the parent's childrearing attitudes and practices, with the supportive parenting style being the best one, encouraging children to express their needs, physical and emotional reactions to the treatment [12].

Little attention has been, as of yet, paid to the caring efforts of parents and if it could be an important moderator in children's adaptation to the daily new life after a diagnosis of cancer. The present study seeks insight into the parenting practices in caring for their child with leukaemia.

*Aims*

The first aim of this study was to understand the parenting behaviours, the coping strategies and adaptability of children with leukaemia along parental narratives and their presence in our participants.

The second aim was to identify if parenting behaviours were different in relation to the child's age. For this purpose, we expected that the supportive parenting style could be an essential tool especially when the child is a pre-schooler and the required tasks include medical procedures (mouth care, conducting physical exams, venepunctures, bone marrow aspirations). Probably, a more necessary and intensive care was needed for infants and toddlers than for older children.

The third aim was to understand if child coping strategies and child adaptability were related to parenting styles and were associated with the child's age. In the literature (i.e., [13]), we have seen that parents' behaviour may affect the child's reactions to treatment and general adjustment in several ways: helping children during medical procedures [14,15], increasing their general compliance with treatment [12,16].

A priority was to develop and test explicit models of how family processes influence the psychological development of children with chronic health conditions. Studies have emphasised the importance of family communication and support [17] and of general family factors in exacerbating or attenuating the impact of the disease on the child [18].

Parental social support may buffer the association between parent and child distress, even if the quality and the timing of support are not clearly established. Another factor linked to distress in children is the quality of family environment [19,20] with cohesive and expressive families more capable of ensuring the adjustment of each family member, and thereby buffering parent and child distress [21].

Based upon the existing literature, we expect that family factors and disease characteristics would be causal independent variables which impact upon children's adaptive behaviours after one year of the child's therapies.

In particular, we think that a general dimension such as Parenting (comprehensive of all the parental strategies to help children to cope with the illness) assessed at the beginning of the child's hospitalisations and treatments, can be an important mediator that impacts upon the child's early coping with medical procedures and upon the child's behavioural adaptation one year post diagnosis.

## 2. Materials and Methods

### 2.1. Procedure

The project was approved by the University of Padua, Psychology faculty with the protocol n. 4039. The families were contacted by a clinical psychologist during the first hospitalisation of their children, about one week after the diagnosis. Project aims were explained, and informed consent was requested. Informal contacts with the participants were kept up on a daily basis, to provide support and motivation for the project. The parents were interviewed in a separate room of the Clinic. The questionnaires were filled in the child's room, with the clinical psychologist's assistance, during a quiet period of the day. The participants were informed that they were free to drop out at any moment of the study. Each family was contacted again at several established time points: 1 month later, 6 months later and 12 months later, but here we will show only the first and the last assessment steps. The assessments were carried out at the Day Hospital or in the library of the Clinic. The psychologist remained constantly in touch with the child and the family for the duration of the study, with frequent telephone contact and direct contact during the DH check-ups. Before the assessments, the psychologist contacted the parent by telephone to agree about the meeting. The data collection phase of the study required several steps. The timing followed two main events: the child's coping and parenting assessment in the second week after the communication of the diagnosis (T1) adopting the EFI-C, and 12 months later (T4) adopting the Vineland Adaptive Behavior Scales (VABS). The first contact took place while the child was under the initial therapy in the Clinic and the next assessment was done when the child and the family came for day hospital visits.

### 2.2. Participants

The patients were 118 children with leukaemia and their parents recruited at the Haematology–Oncologic Clinic of the Department of Child and Woman Health, University of Padua. All parents were Caucasians with a mean age of 37.39 years (SD = 6.03). Most parents had 13 years of school (50.8%); 32.2% had 8 years; 5.9% had a college education; 9.3% had a degree or diploma and 1.7% had 5 years of school. Parents' incomes were average (52.7%), high (24.1%) and low (23.2%) for Italian norms, but above poverty. The average of job hours/weekly were mostly around 35 (28.4%) and 45 (22%), even if the most of parents had partial parental leave or they were housewives (43.4%). The parents who participated were mostly mothers (N = 101) and only a few were fathers (N = 17) because the mothers were more proximal to the child during hospitalisation while fathers stayed with other siblings or continued to work to maintain the family. In the preliminary analysis we controlled for the possible differences between fathers and mothers. There were no

significant differences in our variables so we decided to consider them all together. Children's mean age was 5.89 years (SD = 4.21, range = 1 year–17 years). Mostly children had Acute Lymphoblastic Leukaemia (ALL) (N = 98), while 20 had Acute Myeloid Leukaemia (AML). At the following time-points we had a loss of participants due to several reasons: 11 deceased, 4 changed health centre, 5 relapsed or were in grave illness situation at the assessment moment and only 5 families dropped out from the study (3.9%).

### 2.3. Analyses Plan

Descriptive statistics were used to address the first question. To answer the second question on how the child's age affects parenting behaviour at T1, an ANCOVA was performed with: Parental perception on the child's dimensions of EFI-C (Coping and Adaptability) and demographic factors (parent's school years, parent's mean of job days/a week, parent's Life Stress Events) as covariates; Age of the Child (three levels: 0–3 ys, N = 32; 4–6 ys, N = 45; more than 7 ys, N = 35) as a fixed factor; Parenting as dependent variable.

To answer the second question on which family, child and disease factors just after the diagnosis communication were responsible for long-term adaptation of children with leukaemia after 1 year of treatments, three regression analyses, using Multiple regression for both moderation and mediation effects, were conducted. Baron and Kenny [22] illustrated the three multiple regression analyses testing mediation effects: the significance of the path "predictor A (Parental perception on the child's coping) on the mediator B (Parenting)" was examined in the first regression; the significance of the path "predictor A (Parental perception on the child's coping) on the dependent variable C (VABS global score)" was examined in the second regression; finally, the predictor and mediator used simultaneously as predictors of dependent variable were tested in the last equation.

A regression approach was also used here for testing moderation effects. The predictor and moderator main effects were entered simultaneously into the regression equation first, followed by the interaction of the predictor and moderator. After preliminary analyses of Pearson's bivariate correlations between the several variables tested, a model of mediation effects following the Baron and Kenny's rule (three linear regressions) was performed to evaluate the family factors and disease characteristics at T1 which impacted upon children's adaptive behaviours (Vineland Adaptive Behavior Scales global score) one year after the child's therapies. A post hoc Sobel test was used to control the signification of mediation effects.

### 2.4. Instruments

The EFI-Cancer (EFI-C; [23–25]) is a parent interview which explores the daily routines of family life and the salient concerns regarding how that routine is organised. The interview is a mix of conversation, probing questions by the interviewer and pre-planned questions. Participants use their words and emphases. The interviews start with a question such as: 'Would you guide me through your daily life? What is your and your child's routine'? The EFI interview form flows from our theoretical and epistemological approach, which starts with the observation that the daily family routines and actions constitute adaptation tasks, in which various people participate. Such tasks are carried out, in practice, according to the family resources available and through specific scripts or sets of actions, which are meaningfully linked to the beliefs and values of the broader ecology and culture and which show the emotions and motivations held in them. When the adaptation tasks are overloaded by negative emotion, as often occurred in our case, the participants can sometimes find relief during the interview process [25,26].

A total of 98 items were extracted from the parental narratives and 11 major dimensions were identified, each of them with good internal consistency, three dealing with child experience and eight with the parental one. One-quarter of the total 118 interviews were coded by two independent judges with a score ranging from 0 (low presence of the variable) to 8 (high presence of the variable), showing good Spearman inter-rater reliabil-

ity (rho = 0.833; $p$ = 0.001). The inter-rater agreement could be run only on a part of the interviews because the rho value was really good, and it was not necessary to continue the coding from both the judges. All the remaining interviews were then coded by one judge. We selected only some EFI-C dimensions related to parenting and parental perceptions on the child's coping and quality of life. The other dimensions were related to other topics not important for the aims of this study and therefore were not considered.

### 2.4.1. Vineland Adaptive Behavior Scales (VABS)

The VABS [27,28] are useful in assessing an individual's daily functioning throughout several domains of adaptive functioning (personal and social). In this study we used The Interview Edition, Expanded Form, with 540 items. This form is administered to a parent or caregiver in a semi-structured interview format, and it yields a more comprehensive assessment of adaptive behaviour.

Adaptive behaviours investigated are: Communication, Daily Living Skills, Socialisation and Motor abilities. The Communication domain is comprised of three sub-domains: Receptive, Expressive, and Written Language. The Daily Living Skills scale includes the Personal, Domestic, and Community sub-domains. The Socialisation scale is comprised of the Interpersonal, Play and Leisure, and Coping Skills sub-domains. The Motor scale includes Gross and Fine motor abilities.

Studies confirming the reliability and validity of the VABS have solidified this measure as one of the most widely used assessments of adaptive behaviour. Using a checklist to assess adaptive behaviour may limit the information an interviewer gathers about the various activities involved in an individual's behaviour. A checklist also may allow the respondent to bias the outcome of the assessment because he or she might not fully understand the intent of certain items or might not know the criteria for scoring.

Consciously or unconsciously, the respondent might choose scores that do not reflect the individual's true behaviours. Reliability studies indicate that professionally conducted and scored interviews have higher reliability and validity than checklists. Each item is rated "2" (behaviour is usually or habitually performed), "1" (sometimes or partly performed), or "0" (never performed). In addition, there is a code ("N") for cases when the child has never had the opportunity to perform the activity and a code ("DK") to use when the caregiver does not know if the child performed the activity. In this study we used the raw scores of each domain and of the total adaptation score. Psychometric properties for the Expanded Form in the Italian standardisation version revealed a good reliability and validity and has been adopted also in this clinical population in a precedent study [29].

### 2.4.2. SES Questionnaire

Parental education and occupational status were measured. In particular, the following variables were considered: number of years of school achievement, type and average hours of job, economical status, number of familiars and sons in the family.

## 3. Results

*Parenting Behaviours, Coping Strategies and Adaptability of Children with Leukemia along Parental Narratives*

The first aim was designed to extract the typology and the frequency of parenting behaviours, the child's coping and adaptability from the parental narratives. Table 1 shows the EFI-C dimensions taken into consideration in this study and their descriptive frequencies. We can see that parenting dimension is the higher one, showing a medium–high level, while the parental perceptions of child coping and adaptability placed at a low–medium level.

**Table 1.** EFI-C child and parent dimensions and relative descriptive statistics considered in the present study.

| Dimensions | Items | Mean (Range 0–8) | SD |
|---|---|---|---|
| Parenting the child in the hospital | -creation of links between home and hospital<br>-use of strategies to help the child cope with daily medical procedures<br>-level of trust about leaving the child with others during the day<br>-importance given by the parent about being next to the child during his/her sleep<br>-level of parent–child empathy<br>-perceived parental self-efficacy<br>-perceived ability in soothing the child's cries/desperation<br>-proximity to the child while soothing him/her<br>-perceived difficulties while taking care of the child during hospitalisations | 5.16 | 0.95 |
| Parental perceptions of child coping with procedures and hospitalisation | -acceptance/understanding of the explanations before medical procedures<br>-use of different strategies during medical procedures<br>-monitoring of medical procedures<br>-need for parents before, during, or after medical procedures<br>-need for parents during daily life in the hospital<br>-level of adaptation to the hospital routines<br>-level of adaptation to daily restrictions related to illness<br>-requests for information/reassurance from doctors<br>-coping with painful procedures<br>-tolerance to movement restrictions<br>-level of acceptance of the possible physical changes<br>-coping with emotional stress | 3.94 | 1.36 |
| Parental perception of child adaptability | -level of the child's emotional intensity (crying, anger episodes) associated with specific causes (medical procedures)<br>-sleeping problems<br>-level of the child's general curiosity and attention about the hospital environment related with games and play<br>-level of the child's consolable capacity<br>-capacity of the child to become serene just after medical interventions<br>-parent's perception about the quality of change in the child's relations with doctors and white coats<br>-parent's perception of a stability in the child's characteristics<br>-level of parent's perception of sane aspects of the child in spite of the illness | 5.39 | 1.22 |

The second question addressed in this study was: How does the child's age affect parenting behaviour at T1? To answer this question, an ANCOVA was performed with: Parental perception of the child's dimensions of EFI-C (Coping and Adaptability) and demographic factors (parent's school years, parent's mean of job days/a week, parent's Life Stress Events) as covariates; Age of the Child divided into three levels (0–3 ys; 4–6 ys; more than 7 ys) as a fixed factor; Parenting as dependent variable. As expected, the analysis revealed the presence of a significant group difference on Parenting by Child's Age. When controlling for these independent variables, estimated marginal means for Parenting in the three groups of the child's age were 5.56 (SD = 0.14) for children aged 0–3, 5 (SD = 0.09) for children aged 4–6 and 5.08 (SD = 0.12) for the older children. Results are presented in Table 2.

**Table 2.** ANCOVA comparing parenting between the three categories of the child's age controlling for demographic factors and parental perception of the child's dimensions.

| Source | df | F | $\eta^2 p$ | p | B | t | Sig. |
|---|---|---|---|---|---|---|---|
| Group (child's age): in order 0–3; 4–6; 7≥ | 2 | 4.81 | 0.085 | 0.010 | 0.479 | 2.153 | 0.034 |
| Parent's schooling years | 1 | 0.62 | 0.006 | ns | −0.076 | −0.492 | ns |
| Parent's mean of job days/a week | 1 | 0.07 | 0.001 | ns | | | |
| Parent's life stress events | 1 | 0.09 | 0.001 | ns | | | |
| Parental's perception of the child's coping | 1 | 5.08 | 0.047 | 0.026 | | | |
| Parental's perception of the child's adaptability | 1 | 20.11 | 0.163 | 0.0001 | | | |

The third question was: Which family, child and disease factors just after the diagnosis communication are responsible for long-term adaptation of children with leukemia after 1 year of treatments? To answer this question three regression analyses, following Baron and Kenny's (1986) rules, were used [20]. The significance of the path "predictor Child Coping on the mediator Parenting" was examined in the first regression ($R^2$ = 0.39; β = 0.63; *p* = 0.0001). The significance of the path "predictor Parental perception of child coping on the dependent variable Vineland Adaptive Behavior Scales (VABS) global score" was examined in the second regression ($R^2$ = 0.39; β = 0.62; *p* = 0.0001); finally, Parental perception of child coping (β = 0.69) and Parenting (β = -0.11) were used simultaneously as predictors of VABS global score in the last equation ($R^2$ = 0.40; *p* = 0.0001). Sobel's test was then performed to determine if the relationship between the independent variable and dependent variable has been significantly reduced after inclusion of the mediator variable. The mediation effect was measured by the Sobel test (Z = 4.17; *p* = 0.00003) (Figure 1).

Then, four hierarchical regression models were run to understand better which of the VABS domains were more significantly influenced by parental perception of the child's coping and parenting. Two models obtained a significant prediction of both factors. The first had as dependent variable the Communication domain ($R^2$ = 0.42; *p* = 0.0001) with Parental perception of the child's coping (β = 0.79, *p* = 0.0001) and parenting (β = 0.27, *p* = 0.008) impacting significantly on this. The second had as dependent variable the Motor skills ($R^2$ = 0.28; *p* = 0.001) with Parental perception of the child's coping (β = 0.68, *p* = 0.0001) and parenting (β = 0.40, *p* = 0.001) influencing significantly on these.

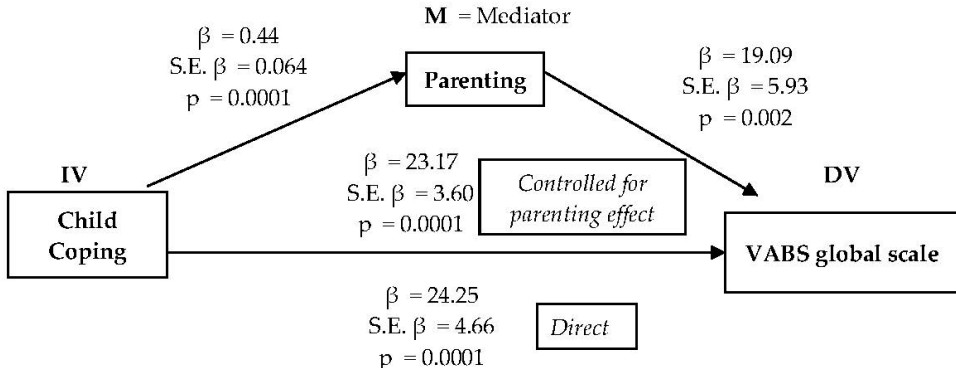

**Figure 1.** Predictor and mediator factors tested at T1 of the VABS global scale tested at T4. Legend: IV = Independent Variable; DV = Dependent Variable; M = Mediator.

## 4. Discussion

Parenting a child with cancer can be very difficult for families, especially when the child is very young and the required tasks include several medical and stressful procedures [10,14]. As expected, the results of the present study confirm the presence of a significant group difference in parenting by the child's age: infants require a higher and more intensive parenting behaviour, followed, respectively, by school-aged children and by pre-school aged children. We think that these findings confirm the difficult role of parents in caring their ill children when they are younger: infants' routines change so much due to hospitalisation in an important developmental time and they need more energy directed towards them. Additionally, the school age can be critical in another sense, because the children comprehend better the situation due to their cognitive maturity. This capacity sometimes is useful but requires intensive parent communication and support. Parents of school-aged children have to manage their parenting functions to help their sons/daughters to tolerate the school and the peer relationship absences. Pre-school children probably have fewer needs in these fields of social relationship and schooling, because they do not yet have many previous routines to maintain, and they require primarily the physical presence of the parent in the hospital daily activities.

To answer the second research question about which family, child and disease factors just after the diagnosis communication are responsible for long-term adaptation of children with leukaemia after 1 year of treatment, we examine the precedent studies, noting that in the literature there is no study that measures which family factors are responsible for the child's long-term behavioural adaptation at specific time points of the illness such as after 1 year. Our related priority is to develop and to test an explicit model of how family processes influence the psychological development of children with chronic disease conditions. Therefore, we expect that some family and children coping and quality of life factors assessed in the early period close to the communication of the diagnosis would be causal independent variables which impact upon children's adaptive behaviours post one year of the child's therapies. In particular, we think that parenting (including of all the parental strategies to help children to cope with the illness) and the child's coping with medical procedures and quality of life assessed at the beginning of the child's hospitalisations and treatments, can impact upon the child's behavioural adaptation one year post diagnosis, but we are interested in understanding if there is a linear causal relationship or a mediation effect between the mentioned factors. Our findings showed that the child's coping with medical procedures at the second week after the diagnosis controlled for parenting effect, impacts upon the child's behavioural adaptation one-year post diagnosis. Early child coping resulted a key factor for the future adaption of the child, but the role of good parenting also reinforced the possibility of the child's adaptive behaviour being good throughout the therapies.

Parenting is a mediator that dampens the direct effect of the child's early coping upon the child's adaptive functioning, entering the model as a key element to be considered. Parenting and parental perception on the child's coping impact on Communication and Motor skills abilities was especially significant. We can appreciate this result because a specific intervention programme can be implemented to help children more at risk just after the diagnosis for developmental delays. Therefore, these results prompt us to think about two parallel programmes: one to increase the child's coping strategies during the first hospitalisation, especially for school-aged children, and the other to teach some parenting strategies to parents of children just at the beginning of therapies, specific for the younger children.

Communication and motor abilities should be enforced adopting, just at the diagnosis time, the child's coping strategies (i.e., distraction, cognitive self-instruction, social support, problem solving). Parenting training could improve the emotional resources of parents for helping children in their adaptive tasks both at the beginning of the illness and after 1 year of therapies, when they could come back to their daily routines such as school or other social activities.

This study has several limitations including the small sample size, the unique centre experience, the use of proxy-report measures without self-reporting from paediatric patients. The strengths of this study are the homogeneity of the diagnosis of children (all leukaemia's), the richness of the qualitative data extracted from parental narratives and the longitudinal design of the study. The longitudinal timing is the authentic strength of the study because until now longitudinal studies about this topic are lacking; therefore, this research filled an important gap.

Recommendations for future research could include: involvement of other Haematology–Oncologic Clinics, the adoption also of self-report measures and standardised tests on the paediatric patients at the different time points and the assessment of efficacy of adoption of intervention programme on parenting or on the child's coping.

**Author Contributions:** Conceptualisation, M.T. (Marta Tremolada) and S.B.; methodology, M.T. (Marta Tremolada) and S.B.; formal analysis, M.T. (Marta Tremolada); investigation and data curation M.T. (Marta Tremolada), resources, S.B.; writing—original draft preparation, M.T. (Marta Tremolada); writing—review and editing, L.T.; visualisation, M.C.P. and M.T. (Manuela Tumino).; supervision, A.B.; project administration, S.B. All authors have read and agree to the published version of the manuscript.

**Funding:** This research was funded by Istituto di Ricerca Pediatrica e Fondazione Città della Speranza; the grant is now expired.

**Institutional Review Board Statement:** The study was conducted according to the guidelines of the Declaration of Helsinki, and approved by the Ethics Committee of the University of Padua, Psychology faculty with the protocol n. 4039.

**Informed Consent Statement:** Informed consent was obtained from all subjects involved in the study.

**Data Availability Statement:** Not Applicable.

**Conflicts of Interest:** The authors declare no conflict of interest. The funders had no role in the design of the study; in the collection, analyses, or interpretation of data; in the writing of the manuscript, or in the decision to publish the results.

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
