# Peer review of "The Moderating Effect of Parenting on Adaptation of Children with Leukemia"

_hemato, doi:10.3390/hemato2020022_

Round 1

Reviewer 1 Report

The study was well planned and discussed. But some inteventions are necessary in my opinion.

I'm surprised at the flow of paragraphs. My expectations were for Introduction, Method with participants, measurements and statistical analysis, and after Results and Discussion. I prefer the traditional flow to the one presented, i.e., Aims, Results, Discussion, Participants, etc., so I suggest this revision that I consider minor, because is sufficient changing paragraph positions to assolve it.

A minor revision is needed on lines 195-197: "four main events" is contradictory with "A first contact" and "five sessions". 

Authors address an important topic: how parenting influences child’
s adaptation to leukemic illness. Especially, two queries are
specifically targeted: does parenting change as a function of
child's age? Can parenting (assessed at the beginning of the
child’s hospitalizations and treatments) be an important mediator
both for child’s early coping to medical procedures and child’s
psychosocial adaptation one year post diagnosis? Results appear to
be convincing because of the adopted method and are well discussed.
The longitudinal timing is the authentic strength of the study. As
Authors note, until now longitudinal studies about this topic are
lacking; therefore, their research filled an important gap.
Despite the strengths, some minor revisions are necessary: first,
it is not specified which domains of the child's adaptation are
affected by parenting one year after diagnosis - in fact, VABS
with their 540 items investigate many repertoires. Second, in the
procedure Authors’ description about "four main events ......"
and "five sessions for the child's assessment" is not clear - it
can be specified better. Finally, I suggest reporting acronyms with their extended
diction - specifically ALL and VABS – at their first use at the
Introduction and Results sections, respectively. »

Author Response

Comments and Suggestions for Authors

The study was well planned and discussed. But some interventions are necessary in my opinion. I'm surprised at the flow of paragraphs. My expectations were for Introduction, Method with participants, measurements and statistical analysis, and after Results and Discussion. I prefer the traditional flow to the one presented, i.e., Aims, Results, Discussion, Participants, etc., so I suggest this revision that I consider minor, because is sufficient changing paragraph positions to assolve it.

Response: We follow the template of the journal as a rule, but we agree with you that it is better to change the paragraph position and we did it. See throughout the text the sections’ disposition order

A minor revision is needed on lines 195-197: "four main events" is contradictory with "A first contact" and "five sessions".

Response: Thank you for your revision. We changed this part clarifying it more. See lines 123-125.

Authors address an important topic: how parenting influences child’s adaptation to leukemic illness. Especially, two queries are specifically targeted: does parenting change as a function of child's age? Can parenting (assessed at the beginning of the child’s hospitalizations and treatments) be an important mediator both for child’s early coping to medical procedures and child’s psychosocial adaptation one year post diagnosis? Results appear to be convincing because of the adopted method and are well discussed.

The longitudinal timing is the authentic strength of the study. As Authors note, until now longitudinal studies about this topic are lacking; therefore, their research filled an important gap.

Despite the strengths, some minor revisions are necessary: first, it is not specified which domains of the child's adaptation are affected by parenting one year after diagnosis - in fact, VABS with their 540 items investigate many repertoires.

Response: We run new hierarchical regression analyses to identify which child’s adaption was more affected significantly by parenting one year after the diagnosis. Communication and motor skills abilities were significantly influenced by child’s coping and by parenting. See new analyses at lines 284-289.

Second, in the procedure Authors’ description about "four main events ......" and "five sessions for the child's assessment" is not clear – it can be specified better. Finally, I suggest reporting acronyms with their extended diction - specifically ALL and VABS – at their first use at the Introduction and Results sections, respectively.

Response: We specified better the acronyms in their first use both in the abstract, in the introduction and in the results sections.

Reviewer 2 Report

Thsi was very difficult to read. I expect some of it was related to the translation, but also the organization left a significant lack of clarity of what was being measured how. After reading the procedures/measures at the end, it was a little clearer (methods should be moved up in the paper, though), but still all teh concepts were all jumbled together. The paper needs to be clearer and more precise about what concepts the authors are talkign about, how it was measured and analyzed. Also, the discussion needs to be much more impactful. The takeaway as written is that infants are harder to parent, and parenting factors are related to how a child copes. I would argue we already knew that. I think this study is important, but it needs rewritten to communicate how and why taht is important, as well as more specific impact (how these results can and will impact clinical practice, and how to do that...).

Author Response

Comments and Suggestions for Authors

This was very difficult to read. I expect some of it was related to the translation, but also the organization left a significant lack of clarity of what was being measured how. After reading the procedures/measures at the end, it was a little clearer (methods should be moved up in the paper, though), but still all teh concepts were all jumbled together.

Response: We changed the sections order as requested. We followed the template of the journal but we reconsidered it along your suggestions. We also checked and revised the translation.

The paper needs to be clearer and more precise about what concepts the authors are talking about, how it was measured and analyzed.

Response: Shifting the method section before it is now clearer that the instruments adopted were the Ecocultural Family Interview from which we extracted factors such as parenting and child’s coping The Child’s adaption was assessed adopting the VABS, that are an in depth and structured interview on parents on children’s development domains

Also, the discussion needs to be much more impactful. The takeaway as written is that infants are harder to parent, and parenting factors are related to how a child copes. I would argue we already knew that. I think this study is important, but it needs rewritten to communicate how and why taht is important, as well as more specific impact (how these results can and will impact clinical practice, and how to do that...).

Response: We ameliorated the discussion along your suggestions and we added the clinical suggestions and the recommendations for future research. See lines 325-339.

Reviewer 3 Report

The stated purpose of this study was to explore “parenting depending on child’s age and to develop and test a model on how family processes influence the psychosocial development of children with leukemia”.. (abstract). As noted in text of manuscript, the study was designed to seek insight regarding parenting practices in caring for a child with leukemia.

 As the authors indicate in the introduction section, having a child with cancer is an overwhelming experience for parents/caregivers.

While the potential significance of this study is recognized,  the logical flow of ideas could be improved.  Specifically, some editing is required. For example, page 3, line 97.. First sentence should state based on the existing literature not basing).

Under section 2 (Results)..first sentence as written is not clear.. ‘The first aim was related to extract from the narratives .. would state.. the first aim designed to extract the typology… from the narratives.  In addition, the results and discussion  sections should come after the Methods and  Materials sections and all included in that section. Normally, results and discussion are last in a manuscript.  The discussion section should place the findings in context of relevant literature and also include limitations of the study and directions for future research in the area of inquiry.

Authors should check manuscript for complete sentences. For example, in Participants section, second paragraph, first sentence.. Caucasian with a mean age..   Please also check verb tenses and be consistent  using past tense for what was actually performed in this study. Page 8 first sentence, for example, For the first question we run descriptive analyses.   Better to state Descriptive statistics were used to address the first question.  While much information is presented on Baron and Kenny’s approach, as presented it is difficult to follow how applied in the current study.

Author Response

Comments and Suggestions for Authors

The stated purpose of this study was to explore “parenting depending on child’s age and to develop and test a model on how family processes influence the psychosocial development of children with leukemia”.. (abstract). As noted in text of manuscript, the study was designed to seek insight regarding parenting practices in caring for a child with leukemia.

As the authors indicate in the introduction section, having a child with cancer is an overwhelming experience for parents/caregivers.

While the potential significance of this study is recognized, the logical flow of ideas could be improved.  Specifically, some editing is required. For example, page 3, line 97.. First sentence should state based on the existing literature not basing).

Response: Thank you for your suggestions. We added the quotation.

Under section 2 (Results)..first sentence as written is not clear.. ‘The first aim was related to extract from the narratives .. would state.. the first aim designed to extract the typology… from the narratives. 

Response: We revised this sentence. Thank you. Lines 240-241.

In addition, the results and discussion  sections should come after the Methods and  Materials sections and all included in that section. Normally, results and discussion are last in a manuscript.

Response: We changed the order, we adopted it following the templates, but we agree with your considerations and we changed it.

 The discussion section should place the findings in context of relevant literature and also include limitations of the study and directions for future research in the area of inquiry.

Response: We revised the discussion along your suggestions. See lines 325-339.

Authors should check manuscript for complete sentences. For example, in Participants section, second paragraph, first sentence.. Caucasian with a mean age..   Please also check verb tenses and be consistent  using past tense for what was actually performed in this study. Page 8 first sentence, for example, For the first question we run descriptive analyses.   Better to state Descriptive statistics were used to address the first question.  While much information is presented on Baron and Kenny’s approach, as presented it is difficult to follow how applied in the current study.

Response: We revised these parts along your suggestions in terms such as Caucasian (line 131) and verb tenses (line 139). We explained better how we applied the Baron and Kenny’s approach (lines 166-168) and erased some too much detail information on this approach (at line 177).

Reviewer 4 Report

First, I like to congratulate the authors for conducting such an interesting study. The title is interesting. However,  the manuscript is not well prepared. Thus, the study leaves the reader with more questions than answers. There are many major concerns that the authors should address 
1) There is no conceptual framework for the study. 
2) Methods of the study were unclear; there is no detail explanation of how the researcher conducted the study (data collection, participant recruitment, questionnaire/instrument used in the study, data analysis) 
3) How researchers measured the parenting behaviours, coping strategies and adaptability of children with leukaemia is not clear. If researchers have used questionnaires to measure all these, what questionnaire were there? Were the questionnaires valid and reliable? 
4) Researcher needs to explain the statistical analysis used in the study and describe the participants' characteristics.    
5) This study requires Human Research Ethnic Committee approval (please attached) 

Author Response

Comments and Suggestions for Authors

First, I like to congratulate the authors for conducting such an interesting study. The title is interesting. However,  the manuscript is not well prepared. Thus, the study leaves the reader with more questions than answers. There are many major concerns that the authors should address

  • There is no conceptual framework for the study.

Response: We added a brief introduction of the conceptual framework for the study. See lines 34-40.

  • Methods of the study were unclear; there is no detail explanation of how the researcher conducted the study (data collection, participant recruitment, questionnaire/instrument used in the study, data analysis)

Response: We changed the order of these sections following also other reviewer’s suggestions and it is now clearer and you can find all this information in the right order.

  • How researchers measured the parenting behaviours, coping strategies and adaptability of children with leukaemia is not clear. If researchers have used questionnaires to measure all these, what questionnaire were there? Were the questionnaires valid and reliable?

Response: We used a narrative approach explained in the method section and we checked also the reliability of the extracted scales. We adopted also a semi-structured interview on parents on child’s adaptation. See lines 178-237.

  • Researcher needs to explain the statistical analysis used in the study and describe the participants' characteristics.

Response: We just put the analyses plan and the participants’ characteristics. See lines 129-177

  • This study requires Human Research Ethnic Committee approval (please attached)

Response: We added in the procedure the code of Ethical committee approval. See line 104

Round 2

Reviewer 2 Report

In the intro, it's not clear how you think coping, parenting, and long-term behaviors are all related.

Procedural questions:

1) Why were only 1/4 of the interviews coded? Please explain

2) In the methods you mention that 11 major dimension were identified, but that doesn't correspond to how any of the things are organized in Table 1. Please explain why you only included 3 dimensions (and 1/4 of interviews).

3) Grammatically, it would be clearer for the reader if you structured the two sentences in lines 263-266 in the same way, so the reader can easily understand. They're the same result, just different dependent variable. But as written, the reader has to work harder to read and understand that.

4) Your results should be more explicit. Did you find mediation according to Baron & Kenny's model? I'm not a mediation expert, but I dont' think you did. Per my review of the mediation methodology, you find mediation when the relationship between the IV and DV goes away, or is reduced, when controlling for the mediator. Your results don't show that...it didn't go away, and in fact the Beta was higher when parenting was controlled in the regression/mediation analysis. I think this warrants some statistical consultation.

5) the conclusions drawn in the discussion are erroneous if mediation was not found. You should rework the discussion to determine exactly what you can interpret from the results and what conclusions you can make. for example: you say in lines 291-294 that parenting can impact coping, but none of your analyses test for that (you do show a relationship between coping and parenting, but your arrow is drawn from coping to parenting; additionally, you don't know what direction that relationship works...does parenting affect coping or coping affect parenting?). In 292-294, you do say that coping, controlling for parenting, impacts one year child behavioral adaptation. Your data do show that, but is that mediation? You haven't shown a mediation effect. Line 295 says that parenting is a mediator that increases the effect of early chidl coping on child behavior. Your stats show that controlling for parenting increases the strength of the relationship between coping and behavior, but that's not mediation. That's the opposite of mediation.

I would recommend revising your discussion based on the data to reflect what you can accurately interpret from your data and results

I don't understand the sentence on 302.

Discussion needs major revision, based on data interpretation.

Author Response

Thank you for your careful revision process. We appreciated it very much.

1) In the intro, it's not clear how you think coping, parenting, and long-term behaviors are all related.

Reply: We add some consideration on this topic. See lines 59-61

Procedural questions:

2) Why were only 1/4 of the interviews coded? Please explain

Reply: The inter-rater agreement could be run only on a part of the participants because the value was really good and it was not necessary to continue the coding for both the judges. We added this part in the text. See lines 191-192.

3) In the methods you mention that 11 major dimension were identified, but that doesn't correspond to how any of the things are organized in Table 1. Please explain why you only included 3 dimensions (and 1/4 of interviews).

Reply: We included all the interviews. ¼ of interviews were reviewed form both the judges, only one judge coded all the interviews. We selected only some EFI-C dimensions related to parenting and parental perceptions on child’s coping and quality of life. The other dimensions were related to other topics not interesting for the aims of this study. We added this part at lines 193-195

4) Grammatically, it would be clearer for the reader if you structured the two sentences in lines 263-266 in the same way, so the reader can easily understand. They're the same result, just different dependent variable. But as written, the reader has to work harder to read and understand that.

Reply: We changed these sentences following your suggestions. See lines 277-280.

5) Your results should be more explicit. Did you find mediation according to Baron & Kenny's model? I'm not a mediation expert, but I dont' think you did. Per my review of the mediation methodology, you find mediation when the relationship between the IV and DV goes away, or is reduced, when controlling for the mediator. Your results don't show that...it didn't go away, and in fact the Beta was higher when parenting was controlled in the regression/mediation analysis. I think this warrants some statistical consultation.

Reply: There is a mistake in the placement of the text-notes in the figure 1. The beta value in the direct effect is higher than that controlled for parenting effect (24.25 versus 23.17). The Sobel test confirms this result. The analysis was run by a statistical expert, there is only a mistake in the figure placement of the notes. Sorry and thank you for your observation.

6) the conclusions drawn in the discussion are erroneous if mediation was not found. You should rework the discussion to determine exactly what you can interpret from the results and what conclusions you can make. for example: you say in lines 291-294 that parenting can impact coping, but none of your analyses test for that (you do show a relationship between coping and parenting, but your arrow is drawn from coping to parenting; additionally, you don't know what direction that relationship works...does parenting affect coping or coping affect parenting?). In 292-294, you do say that coping, controlling for parenting, impacts one year child behavioral adaptation. Your data do show that, but is that mediation? You haven't shown a mediation effect. Line 295 says that parenting is a mediator that increases the effect of early chidl coping on child behavior. Your stats show that controlling for parenting increases the strength of the relationship between coping and behavior, but that's not mediation. That's the opposite of mediation.

Reply: We revised the conclusions with the observed relations. We erased the sentence that parenting impacts on coping, the direction of the regression is coping impacting on parenting.  Instead, we maintained the existence of the mediation effect and the considerations given on this, basing on the statistical consultation and the fact that we revised the mistake in the figure and we added a sentence on it in the results. We changed the sentence that controlling for parenting “increases” - with “dampens”- the strength of the relationship between coping and behaviour. See lines 300-313.

7) I would recommend revising your discussion based on the data to reflect what you can accurately interpret from your data and results. Discussion needs major revision, based on data interpretation.

Reply: It is not necessary to revise the discussion basing on the fact that the mediation effect is present. However, we make some revisions along your precedent suggestions. See lines 300-313

7) I don't understand the sentence on 302.

Reply: We changed this sentence at lines 292-293.

Reviewer 3 Report

The stated purpose of this study was to explore “parenting depending on child’s age and to develop and test a model on how family processes influence the psychosocial development of children with leukemia”.. (abstract). As noted in text of manuscript, the study was designed to seek insight regarding parenting practices in caring for a child with leukemia.

In this  revised manuscript the authors have made concerted efforts to address comments and concerns raised in the initial review.

Several edits remain with the goal of improving the logical flow of content.

Introduction: Lines 44 and 45 repeat lines 34 and 35..  Likely 44 & 45 could be deleted without compromising content.

Page 2: line 76-77: seeks insight into  (add into)

Page 3: line 99.. Based on existing literature

 Line 109: informed consent was requested.. not asked for..

Line 132.. not clear as stated.. even if the major part of parents was temporarily relieves of their work.. ?  Perhaps you are saying  that most had partial parental leave?

Page 6: line 230: The second question addressed IN this study (not from)  affect not affects

Page 8; line 271 and 272;   271: school-aged; 272 pre -school (not scholar)..

274- school age278: same correction school not scholar

Line 307: Limitations not limits..   This study has several limitations including the small sample size, the unique centre experience, the use of proxy-report measures without self-report from pediatric patients.

Line 311: this not their

Author Response

Thank you for your careful revision.

1) Introduction: Lines 44 and 45 repeat lines 34 and 35..  Likely 44 & 45 could be deleted without compromising content.

Reply: Sorry for the mistake. We erased one sentence at lines 44-45.

2) Page 2: line 76-77: seeks insight into  (add into)

Reply: We make this correction, thank you. See line 80

3) Page 3: line 99.. Based on existing literature

Reply: OK, see line 104

4) Line 109: informed consent was requested.. not asked for..

Reply: OK, see line 114

5) Line 132.. not clear as stated.. even if the major part of parents was temporarily relieves of their work.. ?  Perhaps you are saying  that most had partial parental leave?

Reply: OK, see line 136

6) Page 6: line 230: The second question addressed IN this study (not from)  affect not affects

Reply: OK, see line 238

7) Page 8; line 271 and 272;   271: school-aged; 272 pre -school (not scholar)..

274- school age278: same correction school not scholar

Reply: Ok, see lines 285-286 and lines 288 and 292

8) Line 307: Limitations not limits..   This study has several limitations including the small sample size, the unique centre experience, the use of proxy-report measures without self-report from pediatric patients.

Reply: OK, see line 325

9) Line 311: this not their

Reply: Ok, see line 376.

Reviewer 4 Report

I had re-read the entire manuscript. It had been corrected as suggested, thus, I suggest accepting the manuscript. Thank you. 

Author Response

Thank you for your revisions and for your reconsideration. We also used an English proof reading service.  

Round 3

Reviewer 2 Report

Thank you for your revisions which have helped clarify the concerns about mediation. I do think some revision for English grammar would make it more readable, perhaps that is up to the editor to fine-tune.

Reviewer 3 Report

The revised manuscript addresses modifiable changes suggested in prior review. Limitations acknowledged and appropriate directions for future research added.